# Old Drugs, New Battles: Unleashing Repurposed Drug Classes in Triple-Negative Breast Cancer Treatment

**DOI:** 10.3390/ijms262211196

**Published:** 2025-11-19

**Authors:** Vania S. Tshimweneka, Thandi V. Mhlanga

**Affiliations:** Department of Physiology, Faculty of Health Sciences, University of Pretoria, Pretoria 0031, South Africa; u19027011@tuks.co.za

**Keywords:** TNBC, drug repurposing, cancer therapy, FDA-approved drugs, NSAIDs, anti-diabetics, immune modulation, cancer drug resistance, personalized medicine

## Abstract

Cancer remains a major global health challenge, with triple-negative breast cancer (TNBC) representing one of the most aggressive and difficult-to-treat subtypes, characterized by poor prognosis and limited therapeutic options. Current treatments, including chemotherapy, are hindered by high recurrence rates, drug resistance, and severe side effects, highlighting the urgent need for novel therapeutic strategies to address these challenges. Drug repurposing, which involves the application of existing FDA-approved (Food and administration) drugs for new oncological uses, offers a cost-effective and time-efficient alternative to traditional drug development. This review synthesizes recent findings on repurposed drugs, including antidiabetic, antiparasitic, antidepressant, antipsychotic, cardiovascular disease, and non-steroidal anti-inflammatory drugs (NSAIDs), and their potential to target TNBC through mechanisms such as immune modulation, interference with signaling pathways, and inhibition of cancer cell proliferation. Evidence suggests that these agents hold therapeutic promise across heterogeneous TNBC subtypes, although outcomes vary depending on the molecular context. Overall, drug repurposing has emerged as a promising avenue for expanding the treatment options for TNBC; however, further research and personalized approaches are essential to translate these findings into effective clinical applications.

## 1. Introduction

Cancer remains a major global health challenge [1] and is one of the leading causes of death worldwide [2]. In 2020 alone, there were approximately 19.3 million new cases [3] and 10 million deaths [3,4]. Despite current treatments such as chemotherapy, radiation, and surgery [1,5], issues such as severe side effects, drug resistance [5,6], especially in cancer stem cells [7], and treatment failure in advanced stages persist [8,9].

Developing new cancer drugs is costly and time-consuming [3], often taking over a decade and billions of dollars [3,9], with a success rate of less than 1% [3,7]. High drug prices and limited accessibility [8], especially in low- and middle-income countries, further complicate cancer care [2]. Consequently, there is increasing interest in repurposing existing FDA-approved drugs [8], which can significantly reduce development time, cost, and failure rates [5,8].

## 2. Breast Cancer: Triple-Negative Breast Cancer

Breast cancer (BC) is the most prevalent malignancy among women [10] and encompasses multiple molecular subtypes (Figure 1) defined by the differential expression of cell surface receptors [11,12]. The primary intrinsic molecular subtypes include Luminal A, Luminal B, HER2-overexpressing, and basal-like tumors [11,13]. Gene expression profiling has identified triple-negative breast cancer (TNBC) as a subset of basal-like BC, showing a 56% overlap in gene expression patterns [10].

TNBC is an aggressive breast cancer subtype defined by the lack of estrogenic receptor (ER), progesterone receptor (PR), and human epidermal growth factor 2 (HER2) expression [12,13,14]. Its pronounced heterogeneity contributes to varied clinical outcomes, such as early relapse, metastasis, and poor survival [15]. It accounts for approximately 15% to 20% of all BC cases [11,16] and is more prevalent in younger and Black women [10,14]. It has a higher mortality rate than other subtypes, with approximately 40% of advanced-stage patients dying within five years [10,16]. The recurrence rate is approximately 25%, particularly in patients with residual micrometastatic disease following neoadjuvant chemotherapy [16,17]. TNBC frequently metastasizes the brain, lungs, and liver, with a median survival time of 13.3 months for patients with distant metastases [10,17]. Furthermore, TNBC is often associated with *BRCA1* gene mutations, making it a common subtype of hereditary BC [10,16,18]. Molecular profiling has revealed the heterogeneity of TNBC has significant implications for treatment strategies [14]. TNBC has four tumor-specific subtypes including immunomodulatory (IM), luminal androgen receptor (LAR), mesenchymal-like (MES), and basal-like immunosuppressed subtypes 1 and 2 (BL-1 and BL-2) [11,17,18]. The BL-1 subtype of TNBC is characterized by abnormal activation of cell cycle and DNA repair genes, including *MYC*, *PIK3CA*, *BRCA2*, and *TP53*, making it highly responsive to PARP inhibitors and genotoxic agent [11,15,19]. In contrast, the BL2 subtype exhibits dysregulated signaling in pathways such as EGFR, MET, NGF, Wnt/β-catenin, and IGF-1R, with potential treatment strategies involving mTOR inhibitors and growth factor inhibitors [19,20]. The M subtype is marked by high cell motility and differentiation signaling [18], regulated by pathways such as Wnt, TGF-β, and ALK, which contribute to chemotherapy resistance [19]. However, mTOR inhibitors and EMT-targeting drugs may offer therapeutic benefits [19,20]. The IM subtype is enriched in immune response pathways, including T cell receptor signaling, B cell signaling, and IL-12, and is best treated with immune checkpoint inhibitors, such as PD1, PDL1, and CTLA-4 inhibitors [17,20]. The LAR subtype, despite being ER-negative, exhibits high androgen receptor (AR) expression and hormonal pathway activation [16], making anti-AR therapy the most effective treatment [19,20].

Although these subtypes have notable clinical implications [18], their exact impact on patient outcomes remains under investigation, underscoring the need for further research to optimize treatment approaches [10]. Ongoing research into molecular mechanisms, biomarker identification, and targeted therapies is crucial for improving outcomes in patients with TNBC [10,18].

## 3. Overview of Drug Repurposing

Drug repurposing, also known as drug repositioning or reprofiling [3], is a strategy that involves the identification of new therapeutic uses for existing FDA-approved drugs beyond their original indications [3,21]. This innovative approach has garnered attention as a viable alternative for developing anticancer therapies [1,22] and offers significant advantages over traditional drug development processes [8]. Unlike the lengthy and costly traditional journey of developing new drugs, repurposed drugs already possess established safety and efficacy profiles [3], allowing them to bypass the initial phases of clinical trials [21]. Allowing the repurposed drugs to quickly move into Phase II and III clinical trials [8], significantly reducing development timeline to three to five years (Figure 2) and cutting the costs by approximately $300 million [3,8,23]. Drug repurposing generally involves several critical steps, including preclinical studies (in silico, in vitro, and in vivo), clinical observations, and epidemiological data analysis [24]. By understanding a drug’s anticancer effects and its targeted molecular pathways, researchers can identify promising candidates for repurposing. Notably, repurposed drugs may exhibit “off-target” effects, unexpectedly enhancing their antitumor activity. Furthermore, by modulating the gene expression associated with specific cancer profiles, researchers can uncover additional therapeutic possibilities.

Computational methods have become fundamental for identifying drugs suitable for repurposing in cancer therapy, which offers efficient and precise information in predicting novel therapeutic uses for approved compounds [7,25]. Traditional structure-based techniques such as molecular docking and molecular dynamics, alongside signature-based approaches, enable the prediction of drug-target interactions and the identification of compounds capable of reversing disease-associated gene expression patterns [25]. These approaches have evolved to incorporate target-centric, knowledge-driven, pathway-focused, and mechanism-specific models that integrate omics technologies, big data analytics and artificial intelligence to accelerate hypothesis generation and improve candidate selection [7]. Complementary methods, including drug profile-based screening, toxicity-based analyses, and genomics-driven tools such as GWAS and connectivity map further linking drugs to disease-associated genes and biological pathways [26]. Network-based modeling and machine learning frameworks are supported by resources such as Drug Bank. STITCH, and TCGA combining molecular, clinical and pharmacological data to enhance prediction accuracy, reduce experimental costs and support personalized drug repurposing in oncology [27].

This approach has a long history in cancer therapy, dating back to the early chemotherapies derived from mustard gas research [23]. It is estimated that 90% of approved non-cancer drugs, such as antidepressants, anticonvulsants, and statins, can have positive effects when repurposed, either alone or in combination, for cancer therapy [5]. Well-known repurposed drugs include arsenic trioxide, initially approved for external skin conditions, which is now an FDA-approved treatment for acute promyelocytic leukemia (APL) when combined with tretinoin [1,3]. Similarly, thalidomide, originally used to combat morning sickness, has been repurposed for the treatment of refractory multiple myeloma [3,24]. Metformin, an anti-diabetic drug, is also under investigation for its effectiveness against various cancers, including prostate and BC [1,7]. These examples underscore the potential of drug repurposing for developing innovative cancer treatments [3]. Despite these advantages, drug repurposing is not without its challenges [1,23]. Although some repurposed drugs can be used as monotherapies, there is an increased risk of drug resistance [11]. Combination therapies are often more effective because they target multiple oncogenic pathways, but they may lead to increased adverse effects due to drug interactions [1,6]. Therefore, continuous patient monitoring and extensive molecular studies are vital to address these complications [6]. The heterogeneity of cancer responses among patients necessitates personalized treatment approaches, recognizing that the same drug may yield varying results based on individual genetic differences [3,6]. The growing incidence of cancer has also led to a rise in the economic burden on healthcare systems globally [23]. Between 1995 and 2018, cancer drug spending in Europe has more than tripled, and the global cancer drug market is projected to reach $377 billion by 2027 [9,23]. Given the high cost and low approval rates of novel cancer drugs, repurposing offers a practical solution for reducing costs and improving access to effective treatment [9].

Several drug classes, including anti-diabetic agents, antidepressants, antifungals, antiparasitic, antipsychotics, cardiovascular disease drugs, and NSAIDs, have emerged as promising candidates in cancer therapy research [3]. These drugs are being explored for their ability to interfere with cancer cell proliferation, angiogenesis, immune evasion and metabolic dysregulation. They are currently undergoing various stages of research and require further investigation, particularly in light of the key findings that have already emerged (Table 1). This review examines the advancements in repurposing these drugs for TNBC treatment, highlighting their mechanisms of action and therapeutic potential.

## 4. Repurposed Drugs for TNBC Therapy

### 4.1. Anti-Diabetic Drugs for TNBC Therapy

The growing recognition of the relationship between metabolic disorders, particularly diabetes, and cancer has sparked interest in repurposing antidiabetic drugs for cancer treatment [28]. Both conditions share common risk factors, such as obesity and aging, which contribute to insulin resistance, chronic inflammation, and an environment conducive to tumor development [29,30]. Molecular pathways, including insulin signaling, AMPK, and mTOR, are implicated in both diabetes and cancer, linking the two diseases further [31]. Diabetes can influence cancer progression through mechanisms such as hyperinsulinemia, hyperglycemia, and inflammation, with elevated insulin and insulin-like growth factors (IGFs) promoting cell proliferation and tumorigenesis [30]. While long-acting insulins, such as insulin glargine, have been associated with an increased risk of pancreatic and colorectal cancers, the evidence regarding the impact of insulin treatment on cancer outcomes remains mixed [28], particularly in patients with BC [28,31]. The complex nature of diabetes treatment, which often requires a combination of medications, complicates the assessment of the effects of individual drugs on cancer risk [29]. Due to the increasing prevalence of type 2 diabetes, studies predominantly involve patients with this condition [28,29], yet the distinct hormonal and metabolic profiles of type 1 diabetes necessitate further investigation [29]. Despite the challenges in current research, emerging evidence suggests that specific antidiabetic drugs may offer protective effects against various cancers [30], prompting further exploration of their therapeutic potential in oncology.

Antidiabetic drugs, particularly metformin, exert anticancer effects in TNBC through multiple mechanisms. Metformin inhibits mitochondrial complex I, leading to reduced ATP production, activation of AMP-activated protein kinase (AMPK), and subsequent suppression of the mTOR pathway, which is crucial for TNBC cell proliferation [32] (Figure 3). Additionally, metformin reduces lipid metabolism by targeting fatty acid synthase (FASN) and cholesterol biosynthesis, which are essential for TNBC growth [33]. It also decreases the expression of glucose transporters (GLUT1) in TNBC cells, thereby limiting glucose uptake and energy supply [32,33]. Furthermore, metformin impairs cancer stemness and inhibits epithelial-to-mesenchymal transition (EMT), thereby reducing TNBC’s metastatic potential [28].

Preclinical studies have demonstrated that metformin effectively reduces TNBC cell proliferation and tumor growth in vivo. In TNBC xenograft models, metformin significantly decreased tumor size by altering metabolic pathways [30]. Observational studies have also shown that patients with diabetes receiving metformin have a lower incidence of TNBC and improved survival rates [32]. However, clinical trials evaluating metformin in patients with non-diabetic TNBC have shown mixed results, suggesting that metabolic status may influence drug efficacy [28]. Metformin has shown strong potential in combination with standard TNBC treatments. It enhances the effectiveness of chemotherapy by sensitizing TNBC cells to DNA-damaging agents such as doxorubicin and cisplatin [34]. Additionally, metformin has been reported to improve the efficacy of immune checkpoint inhibitors by modulating the tumor microenvironment and reducing immunosuppressive signals [30]. It also helps overcome drug resistance by targeting cancer stem cells and inhibiting adaptive metabolic pathways that promote TNBC survival under treatment stress [32].

Metformin has a well-established safety profile, with mild and manageable side effects such as gastrointestinal discomfort [28]. Unlike cytotoxic chemotherapy, metformin is associated with minimal systemic toxicity, making it a suitable candidate for long-term use in cancer therapy [32]. However, potential risks, such as lactic acidosis in patients with renal impairment, must be carefully monitored [30,34]. One major challenge with metformin is its low bioavailability owing to poor intestinal absorption and reliance on organic cation transporters (OCTs) for cellular uptake [28]. The limited permeability of metformin in tumor tissues reduces its direct anticancer effects, prompting research into nanoparticle-based formulations to improve drug delivery [32]. Other antidiabetic drugs, such as thiazolidinediones (TZDs), also face bioavailability challenges, requiring dose optimization for effective treatment of TNBC [30,33].

Repurposing antidiabetic drugs, such as metformin, for TNBC presents a promising and cost-effective strategy with metabolic and immunomodulatory advantages. While preclinical and observational studies have shown encouraging results, further clinical trials are necessary to optimize treatment protocols, address bioavailability challenges, and identify patient populations most likely to benefit from metformin-based TNBC therapy.

### 4.2. Antifungal Drugs for TNBC Therapy

Antifungal drugs have shown promising potential for treating TNBC because of their ability to target key oncogenic pathways. Imidazole antifungals, such as ketoconazole and clotrimazole, exert anticancer effects by inducing apoptosis and cycle arrest, particularly in TNBC cells [35]. These drugs inhibit matrix metalloproteinase 9 (MMP9) as seen in Figure 4, reducing TNBC invasiveness and metastatic potential [35,36]. Additionally, itraconazole has been identified as a potent inhibitor of the Hedgehog signaling pathway, which is frequently dysregulated in TNBC, contributing to tumor growth and chemoresistance [37]. Itraconazole targets the SMO smoothened; by inhibiting the SMO it suppresses the downstream nuclear translocation of GLI1 and other GLI proteins that aid in the gene expression program that promotes TNBC [38].

Preclinical studies have demonstrated that antifungal drugs effectively suppress TNBC proliferation and metastasis both in vitro and in vivo. Ketoconazole, for instance, has been shown to inhibit tGLI1, a key transcription factor involved in breast cancer brain metastasis [39]. Similarly, itraconazole, alone or in combination with rapamycin, has been found to induce G_0_/G_1_ phase arrest in TNBC cells, significantly reducing their proliferative capacity [40]. While several observational and retrospective studies suggest a correlation between antifungal drug use and improved cancer outcomes, large-scale clinical trials are still needed to validate their efficacy in TNBC patients [36,41]. The combination of antifungal agents with standard TNBC therapies yields promising results. Itraconazole enhances the effects of chemotherapy by inhibiting drug efflux transporters, thereby overcoming multidrug resistance [41]. Additionally, ketoconazole sensitizes TNBC cells to radiation therapy by modulating DNA repair pathways [39]. These findings suggest that antifungal drugs may serve as valuable adjuncts to conventional TNBC treatments [36].

Despite their therapeutic potential, the pharmacokinetic properties of antifungal drugs pose challenges. For example, itraconazole has poor solubility and bioavailability, limiting its systemic effectiveness [37]. To address this issue, nanoparticle formulations and lipid-based carriers have been explored to enhance drug delivery and tumor targeting [40,41]. The safety profiles of antifungal drugs are well established owing to their long-term clinical use in the treatment of fungal infections. However, some antifungal agents, such as ketoconazole, are associated with hepatotoxicity, necessitating careful monitoring in patients with cancer [39]. Further clinical studies are required to optimize their efficacy and safety in aggressive breast cancer subtypes [41].

### 4.3. Antiparasitic Drugs for TNBC Therapy

Anthelmintic drugs, initially developed for treating parasitic infections, are increasingly recognized for their potential in cancer therapy because of their diverse mechanisms of action [8]. These drugs have been extensively studied for their pharmacokinetic and pharmacodynamic properties and toxicity profiles [42]. Their side effects are well-established and generally acceptable [3,8]. This understanding has led researchers to explore the potential of repurposing these agents for cancer treatment, as some exhibit antitumor properties similar to their antiparasitic effects [42,43]. These agents have been shown to target various cancer-related pathways, including cell cycle regulation and apoptosis induction, making them promising candidates for repurposing in oncology [3,43]. These drugs have shown promise in the treatment of TNBC by targeting oncogenic pathways and overcoming resistance mechanisms.

Mebendazole, a benzimidazole-class antiparasitic drug, exerts anticancer effects by inhibiting tubulin polymerization, leading to mitotic arrest and apoptosis in TNBC cells [44] (Figure 5). Additionally, mebendazole modulates the Wnt/β-catenin signaling pathway, which is crucial for TNBC stemness and metastasis, thereby reducing tumor aggressiveness [45]. Ivermectin, another repurposed antiparasitic drug, targets TNBC cells by inhibiting the Akt/mTOR pathway, inducing apoptosis, and impairing TNBC cell migration [46].

Preclinical studies have demonstrated the efficacy of antiparasitic drugs against TNBC both in vitro and in vivo [47]. Mebendazole has significantly reduced tumor growth in TNBC xenograft models, improving survival outcomes in mice [44,47]. Similarly, ivermectin has been reported to suppress TNBC proliferation and enhance chemosensitivity in resistant cancer cells [42]. Despite these promising findings, clinical trials evaluating the direct impact of these drugs on patients with TNBC remain limited [45]. The combination of antiparasitic drugs with existing TNBC therapies has shown potential for overcoming drug resistance in TNBC. Mebendazole enhances the cytotoxic effects of paclitaxel by disrupting microtubule stability, thereby sensitizing TNBC cells to chemotherapy [22,44]. Ivermectin has been found to work synergistically with immune checkpoint inhibitors by modulating tumor immunogenicity [46], thereby increasing the efficacy of immunotherapy. These findings suggest that antiparasitic drugs could serve as effective adjuncts to standard TNBC treatments [3,48].

The safety profiles of antiparasitic drugs are well established owing to their long-term clinical use in treating parasitic infections. Mebendazole is generally well tolerated, with mild gastrointestinal side effects, whereas ivermectin has been associated with rare neurotoxicity at high doses [44]. Despite their potential, the pharmacokinetics and bioavailability of antiparasitic drugs pose challenges in TNBC treatment [22,45]. Mebendazole has poor water solubility and limited systemic absorption, necessitating the development of nanoparticle-based formulations to improve its delivery [45]. Ivermectin, although well-absorbed orally, has limited tumor penetration, which may require alternative administration routes or combination strategies [42].

Using these antiparasitic drugs will reduce costs and present a promising therapeutic strategy; however, further clinical investigations are needed to optimize their efficacy and safety in this aggressive breast cancer subtype [48].

### 4.4. Antidepressant and Antipsychotic Drugs for TNBC Therapy

Antipsychotic medications, traditionally used to manage psychiatric disorders such as schizophrenia and bipolar disorder, are increasingly recognized for their potential antitumor effects [3,49]. This growing interest is driven by the rising incidence of drug resistance in cancer therapies and the prohibitive costs associated with developing new anticancer drugs [50]. Depression is a common complication in patients with cancer, with a prevalence significantly higher than that in the general population (12.5% vs. 3.3%) [51]. Its presence can severely impact quality of life, treatment adherence [52], and overall mortality risk, contributing to a 39% increase in cancer mortality [51]. Consequently, antidepressants are frequently used alongside cancer therapies to alleviate depressive symptoms and potentially enhance treatment outcomes [53].

Research indicates that these psychotropic agents target multiple hallmarks of TNBC, including immune modulation, resistance mechanisms, apoptosis induction, metabolic disruption, and modulation of various signaling pathways that are critical in tumor development [50,54]. The anticancer effects of these drugs are mediated through diverse mechanisms. Selective serotonin reuptake inhibitors (SSRIs), such as fluoxetine and paroxetine, have been shown to induce apoptosis in TNBC cells by targeting the PI3K/AKT/mTOR signaling pathway, which is critical for cell survival and proliferation [55]. Tricyclic antidepressants (TCAs), including imipramine, can block DNA repair mechanisms and inhibit estrogenic receptor signaling, effectively sensitizing TNBC cells to PARP inhibitors [56]. Phenothiazine-based antipsychotics, such as thioridazine, have been identified as potent inhibitors of drug efflux pumps, thereby enhancing the intracellular retention of chemotherapeutic agents and overcoming multidrug resistance [56,57].

Preclinical studies have provided substantial evidence supporting the efficacy of these drugs in TNBC models. Paroxetine enhances the cytotoxic effects of standard chemotherapy agents, such as doxorubicin and carboplatin, leading to increased apoptosis in TNBC cells [55]. Similarly, imipramine has been evaluated in TNBC patient-derived xenograft models, where it significantly reduced tumor burden and improved survival [56,58]. Clinical trials investigating the role of SSRIs and TCAs in breast cancer treatment are ongoing, though more data are needed to confirm their therapeutic potential [51,59]. One of the most compelling aspects of these drugs is their synergistic potential with existing therapies for TNBC. By modulating autophagy and apoptosis pathways, antidepressants such as fluoxetine and imipramine enhance the efficacy of chemotherapy and targeted treatment [60]. Moreover, their ability to inhibit multidrug resistance pumps suggests that they could improve the bioavailability of conventional TNBC therapies, addressing one of the key challenges in treatment resistance.

Pharmacokinetically, these drugs exhibit favorable absorption and distribution properties, with many demonstrating good blood–brain barrier penetration, which could be beneficial in preventing TNBC brain metastases [56]. However, issues such as drug metabolism and formulation challenges must be addressed to optimize their efficacy in oncology. The safety of these agents has been well-established from their long-term use in psychiatric disorders. However, dose adjustments may be required to minimize potential toxicities when repurposed for cancer therapy [55].

The repurpose of antidepressants and antipsychotics represents a promising strategy for TNBC treatment, with the potential to enhance current therapeutic approaches by targeting key resistance mechanisms and modulating tumor cell metabolism. Further clinical evaluation is necessary to establish their role in routine oncological care and optimize their integration into treatment regimens for TNBC.

### 4.5. Cardiovascular Disease Drugs for TNBC Therapy

Cardiovascular drugs have garnered significant attention for their potential repurposing in the treatment, given their well-established safety profiles and wide clinical use. Several drug classes, including β-blockers, angiotensin-converting enzyme inhibitors (ACE-Is), angiotensin II receptor blockers (ARBs), statins, and aspirin, have been evaluated for their anticancer properties.

β-Blockers exert anticancer effects by inhibiting β-adrenergic signaling, which is implicated in TNBC progression by promoting proliferation, metastasis, and angiogenesis [61]. Propranolol, a non-selective β-blocker, has been shown to reduce TNBC cell proliferation, migration, and invasion in vitro, while in vivo studies have demonstrated the inhibition of tumor growth and metastasis [61,62]. The mechanism involves the modulation of PI3K/AKT and MAPK signaling pathways, suppression of vascular endothelial growth factor (VEGF) expression (Figure 6), and enhanced immune response through the restoration of natural killer cell activity [61]. Similarly, ACE-Is and ARBs have been investigated for their ability to modulate the tumor microenvironment. This is achieved by inhibiting the renin–angiotensin system (RAS). These drugs can downregulate VEGF, MMPs, and pro-inflammatory cytokines, leading to reduced angiogenesis, tumor growth, and metastasis [34].

Preclinical studies have supported the efficacy of these cardiovascular drugs in TNBC models. β-blockers, such as propranolol, demonstrate significant tumor-inhibitory effects, and ACE-Is/ARBs have shown tumor regression in xenograft models [61]. Statins, known for their cholesterol-lowering properties, also exhibit anticancer activity by inhibiting the mevalonate pathway, which is crucial for the prenylation of oncogenic proteins, such as Ras and Rho [62]. This disruption results in reduced proliferation and enhanced apoptosis in TNBC cells [62,63]. However, the clinical data on this is limited [64]. Observational studies suggest potential survival benefits for patients with TNBC using β-blockers in combination with chemotherapy; however, randomized controlled trials (RCTs) are needed for conclusive evidence [61]. Synergistic effects with existing TNBC therapies have been observed for some cardiovascular drugs [64]. For instance, propranolol has been reported to enhance the efficacy of chemotherapy, increasing the survival of animals with TNBC treated with paclitaxel or fluorouracil [61]. Metformin, an antidiabetic drug with cardiovascular benefits, enhances the response to chemotherapy by inhibiting the PI3K/AKT/mTOR pathway and reducing resistance to cytotoxic agents [34].

The pharmacokinetic properties of these drug classes vary. β-Blockers and ACE-Is are well absorbed and widely distributed but may have limited tumor penetration. Statins, particularly lipophilic statins, demonstrate better tumor uptake, which may enhance their anticancer efficacy [62]. However, drug formulation challenges, such as the need for targeted delivery, remain a hurdle for optimal repurposing in TNBC [64]. The safety profiles of cardiovascular drugs are well documented owing to their extensive use in non-oncological conditions. β-blockers and ACE-Is are generally well tolerated, with manageable side effects such as hypotension and fatigue [62,64]. Statins can cause myopathy and hepatotoxicity; however, these risks are considered acceptable in the context of their potential anticancer benefits [62,63]. Nevertheless, the lack of RCTs evaluating the safety of these drugs in TNBC patients highlights the need for further clinical investigation [61].

While cardiovascular drugs show promise in TNBC treatment, particularly in combination with standard therapies, their clinical validation remains a challenge. Future research should focus on well-designed clinical trials to confirm their efficacy, optimize dosing strategies, and evaluate the long-term safety in patients with TNBC.

### 4.6. NSAIDs for TNBC Therapy

Nonsteroidal anti-inflammatory drugs (NSAIDs) have gained significant attention for their anticancer properties [65,66] and play a role in both chemoprevention and chemotherapy [67]. While traditional theories link their effects primarily to the inhibition of cyclooxygenase-2 (COX-2) and reduced prostaglandin production [68], emerging research highlights COX-independent mechanisms, where NSAIDs interact with various proteins, enhancing their anticancer activities [65,69] and helping prevent deep vein thrombosis (DVT) in cancer patients [70]. Evidence indicates that long-term, low-dose NSAID use is associated with improved cancer prognosis and reduced cancer-related mortality, leading to interest in their potential to inhibit metastasis [67,70].

These drugs exert anticancer effects primarily by inhibiting COX-2, which is overexpressed in TNBC and contributes to tumor growth, immune evasion, and metastasis [63]. NSAIDs, such as aspirin, indomethacin, and naproxen, induce apoptosis, reduce angiogenesis, and inhibit the EMT, which is essential for metastasis [71]. Some NSAIDs also exert COX-independent effects, including modulation of the NF-κB and PI3K/AKT pathways, further enhancing their anticancer potential [72].

Preclinical studies have demonstrated the efficacy of NSAIDs in TNBC models in vivo. Indomethacin and naproxen have been shown to reduce TNBC cell proliferation, induce apoptosis, and inhibit migration in vitro [63]. In vivo, these drugs suppress tumor growth and enhance chemosensitivity in animal models. Clinical studies have provided mixed results, with some observational studies suggesting a reduced risk of breast cancer recurrence among NSAID users [71]. However, more extensive RCTs are needed to establish their definitive role in TNBC therapy [73]. NSAIDs also demonstrate synergistic potential with existing therapies for TNBC. Studies have suggested that COX-2 inhibition enhances the effectiveness of chemotherapy and targeted agents by reducing inflammation-driven resistance mechanisms [24]. Specifically, NSAIDs can increase the sensitivity of TNBC cells to paclitaxel and doxorubicin, potentially improving the treatment outcomes [72].

Regarding pharmacokinetics, NSAIDs generally exhibit good oral bioavailability and widespread tissue distribution [63]. However, their short half-lives and gastrointestinal side effects present challenges for their long-term use in oncology. Some formulations, such as nano-formulated NSAIDs, are being explored to improve delivery and reduce toxicity [72]. The safety profile of NSAIDs is well established owing to their extensive use in pain management and inflammatory conditions. However, long-term use may lead to gastrointestinal ulceration, cardiovascular risk, and renal impairment [72]. The balance between therapeutic benefits and potential toxicities must be carefully evaluated before widespread adoption in TNBC treatment [24,72].

NSAIDs hold promise as adjuvant agents in TNBC therapy because of their anti-inflammatory, pro-apoptotic, and chemo-sensitizing properties. Further clinical investigations are required to optimize their use, determine appropriate dosing, and evaluate their impact on patient outcomes in TNBC patients.

## 5. Conclusions

Repurposing FDA-approved drugs for cancer treatment offers a promising and cost-effective approach by leveraging their established safety profiles and expediting clinical trials. Drug classes such as antidepressants, antidiabetics, antiparasitic, and antipsychotics have shown potential in targeting cancer-related pathways, inducing apoptosis, and enhancing standard treatments. Similarly, NSAIDs exhibit anticancer properties by modulating malignancy growth, immune responses, and cell survival mechanisms. While drug repurposing holds great promise for improving cancer outcomes, challenges such as drug resistance, toxicity, and dosage optimization remain. Ongoing research and clinical trials are essential to validate these therapies and refine their integration into oncology.

## Figures and Tables

**Figure 1 ijms-26-11196-f001:**
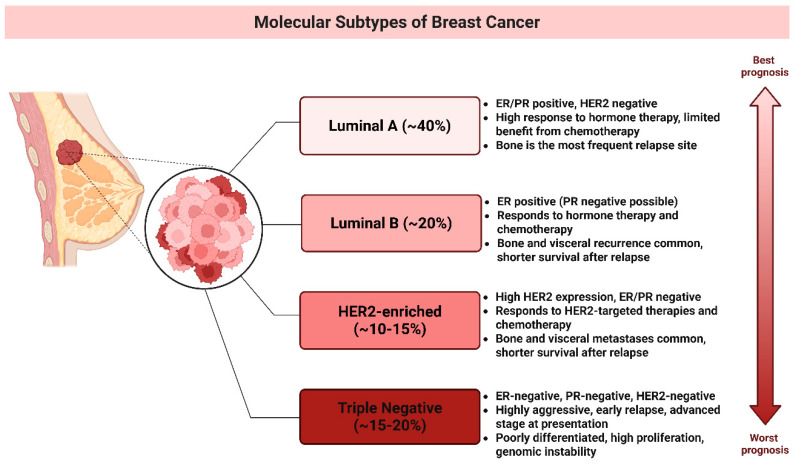
Breast Cancer Subtypes and Their Prognoses. This schematic categorizes breast cancer into four primary molecular subtypes based on gene expression profiles and clinical characteristics: Luminal A (~40%), Luminal B (~20%), HER2-enriched (~10–15%), triple-negative (~15–20%), and a general category. The vertical arrow illustrates the prognosis gradient from best (Luminal A) to worst (TNBC). (Image created in Biorender. VS Tshimweneka. (2025) https://app.biorender.com/illustrations/682eb8d751f88b68ee1c2363?slideId=43839ab2-b059-4c64-ac64-4d1a654419ce and Microsoft PowerPoint 2016 (2016 Microsoft Corporation, Redmond, WA, USA).

**Figure 2 ijms-26-11196-f002:**
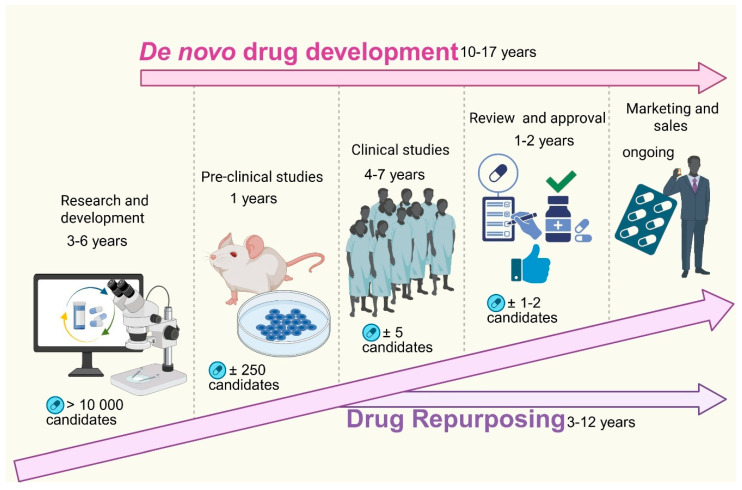
Comparison of de novo drug development and drug repurposing timelines and processes. The figure illustrates the sequential stages and timeframes involved in traditional de novo drug development compared to drug repurposing. De novo drug development, which takes approximately 10–17 years, begins with research and development (3–6 years), during which over 10,000 potential candidates are screened. This is followed by pre-clinical studies (1 year) involving approximately 250 candidates, clinical studies (4–7 years) narrowing down to around five candidates, and finally regulatory review and approval (1–2 years) for one to two viable drugs before entering the marketing and sales phase. In contrast, drug repurposing significantly reduces development time (3–12 years) by leveraging existing drugs with known safety profiles, bypassing early discovery phases, and often expediting entry into clinical trials and approval, thereby accelerating patient access to new treatments. (Image created in Biorender. VS Tshimweneka. (2025) https://app.biorender.com/illustrations/682eb8d751f88b68ee1c2363?slideId=43839ab2-b059-4c64-ac64-4d1a654419ce).

**Figure 3 ijms-26-11196-f003:**
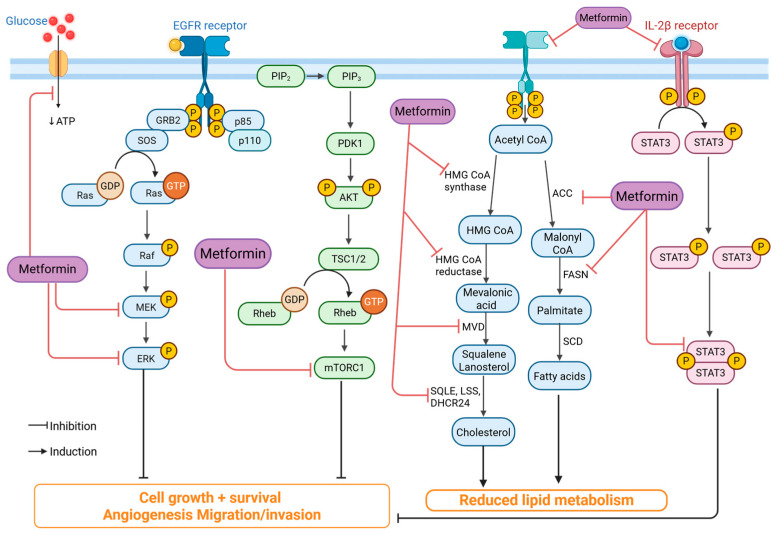
Mechanism of metformin’s anticancer activity. Metformin inhibits key oncogenic pathways, including EGFR/Ras/MEK/ERK, PI3K/AKT/mTOR, and IL-2β/STAT3, leading to reduced cell proliferation, angiogenesis, and invasion. It also suppresses lipid and cholesterol synthesis by downregulating enzymes such as HMG-CoA reductase, ACC, and FASN. Thereby limiting tumor growth and metastasis. (Image created in Biorender. VS Tshimweneka. (2025) https://app.biorender.com/illustrations/682eb8d751f88b68ee1c2363?slideId=43839ab2-b059-4c64-ac64-4d1a654419ce).

**Figure 4 ijms-26-11196-f004:**
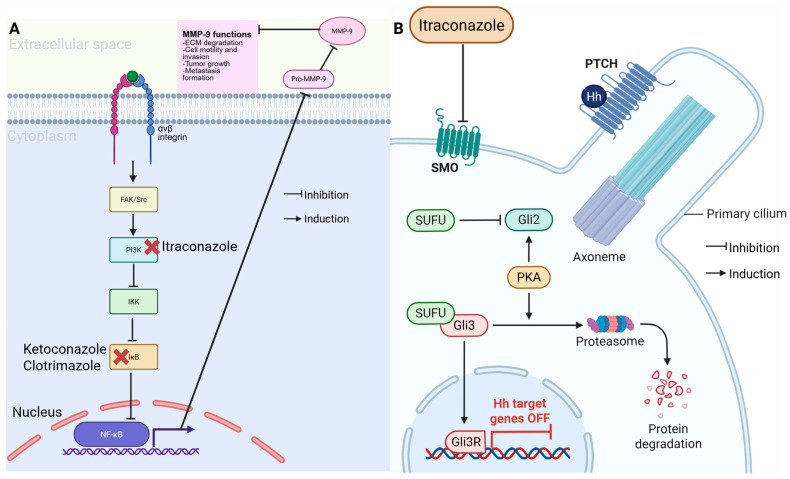
Anticancer mechanism of azole antifungals. (**A**) Itraconazole inhibits the PI3K/NF-κB pathway, reducing MMP-9 activity and limiting cell invasion and metastasis. Ketoconazole and clotrimazole similarly block IκB activation. (**B**) Itraconazole also suppress Hedgehog (Hh) signaling by inhibiting SMO, preventing GLI activation and turning off Hh target genes, thereby reducing tumor growth. (Image created in Biorender. VS Tshimweneka. (2025) https://app.biorender.com/illustrations/682eb8d751f88b68ee1c2363?slideId=43839ab2-b059-4c64-ac64-4d1a654419ce.)

**Figure 5 ijms-26-11196-f005:**
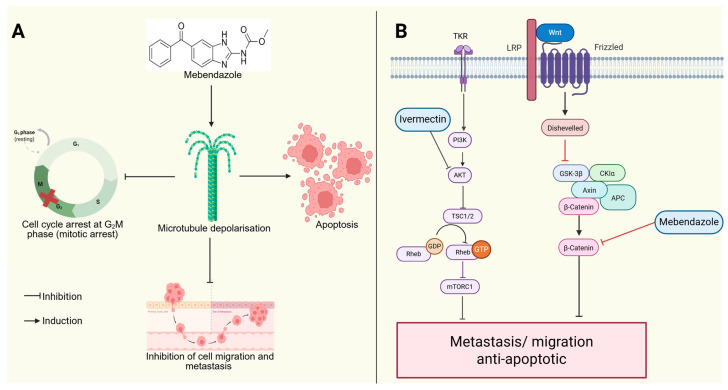
Anticancer activity of mebendazole and ivermectin. (**A**) Mebendazole induces microtubule depolarization, causing G_2_/M phase arrest, which stops the cell cycle from progressing to the next phase (shown by the red cross), triggers apoptosis, and inhibits cell migration and metastasis. (**B**) Mebendazole also blocks the Wnt/β-catenin signaling, while ivermectin inhibits the PI3K/AKT/mTOR pathway and together they reduce metastasis and promote apoptosis. (Image created in Biorender. VS Tshimweneka. (2025) https://app.biorender.com/illustrations/682eb8d751f88b68ee1c2363?slideId=43839ab2-b059-4c64-ac64-4d1a654419ce).

**Figure 6 ijms-26-11196-f006:**
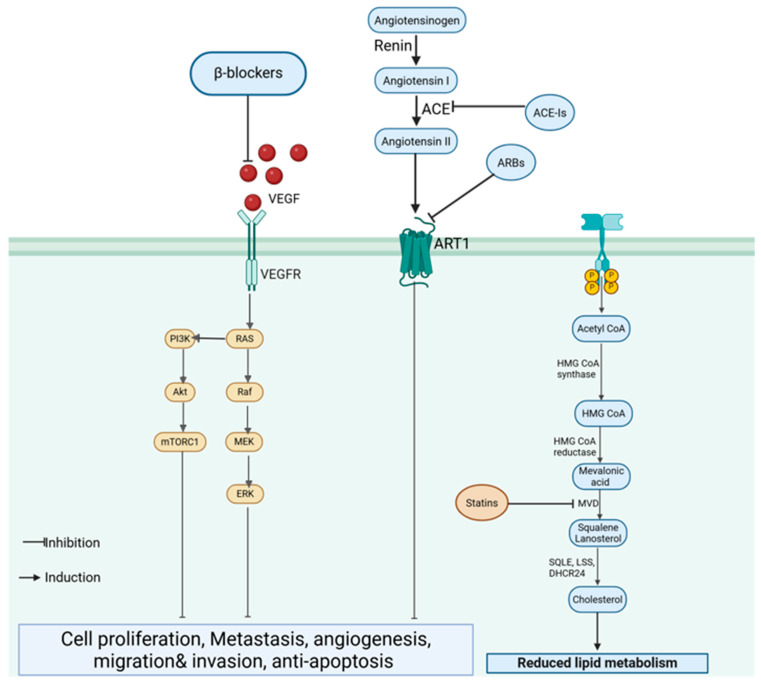
Cardiovascular drugs targeting TNBC progression. β-blockers inhibit VEGF signaling and downstream PI3K/AKT and MAPK pathways, while ace-Is and ARBs suppress the renin–angiotensin system (RAS). Statins block the mevalonate pathway, reducing lipid metabolism, angiogenesis and tumor growth. (Image created in Biorender. VS Tshimweneka. (2025) https://app.biorender.com/illustrations/682eb8d751f88b68ee1c2363?slideId=43839ab2-b059-4c64-ac64-4d1a654419ce).

**Table 1 ijms-26-11196-t001:** Summary of completed and ongoing trial investigating repurposed drugs for TNBC. The table highlights drug classes, trial phases, study status, patient populations or treatment combinations and key outcomes. The drugs demonstrate varying degrees of preclinical and clinical success reflecting the growing potential of drug repurposing strategies in TNBC therapy. (Image was designed by VS Tshimweneka using Microsoft Word 2016 (2016 Microsoft Corporation, USA)).

Drug	Phase	Status	Population/Combination	Key Outcomes
**Metformin**	Phase III	Registered	Breast cancer patients (studied as anticancer adjuvant; evaluated effect on tumor growth/recurrence).	Large, randomized study evaluating metformin’s anti-tumor potential in breast cancer; mixed results reported across trials and further combination approaches under investigation.
**Propranolol**	Phase II	Active/listed as a current phase II study	Metastatic/unresectable PD-L1+ TNBC in combination with pembrolizumab and chemotherapy.	Designed to test tumor re-sensitization; strong preclinical rationale and prior window-of-opportunity propranolol studies in breast cancer.
**Propranolol (neoadjuvant window)**	Phase II (window design)	Completed/reported as conducted	Early breast cancer (window between diagnosis and surgery) to test biological effects of short-term propranolol.	Window studies showed reduced proliferation markers; supports further combination trials in TNBC.
**Ivermectin**	Phase II	Registered/ongoing investigator-initiated studies; check record for recruitment status.	Metastatic triple-negative breast cancer ivermectin combined with PD-1/PD-L1 agents.	Preclinical data show ivermectin can induce immunogenic cell death and enhance T-cell infiltration; trial assesses safety/dose and efficacy signals.
**Mebendazole**	Early-phase/pilot clinical study (various cancer types)	Registered (some trials in other advanced cancers); TNBC-specific clinical data limited and strong preclinical TNBC evidence.	Advanced solid tumors (some trials include GI or unknown primary); preclinical TNBC/xenograft data show efficacy, incl. CNS metastasis models.	Promising preclinical TNBC results (xenografts, CNS metastasis models); clinical trials in cancer patients exist but TNBC-specific clinical results are still limited.
**Itraconazole**	Early clinical use/pilot reports	Published small clinical reports; some investigator studies showed responses in heavily pretreated patients.	Metastatic breast cancer (including some TNBC cases) is often combined with chemotherapy/rapamycin in preclinical models.	Some clinical reports suggest benefits in heavily pre-treated cases; larger RCTs lacking—strong preclinical rationale (Hedgehog/PI3K inhibition).
**Statins**	Observational/exploratory clinical analyses	Observational data and institutional analyses; clinical RCTs limited	Concurrent statin uses in patients treated for TNBC (evaluated for survival outcomes/response)	Multiple observational reports suggest improved survival or reduced recurrence with statin use in TNBC; RCT evidence limited, and trials suggested.
**NSAIDs (aspirin, indomethacin, naproxen)**	Mostly observational; some small interventional efforts	Mixed evidence from observational cohorts; RCTs limited for TNBC-specific endpoints	Long-term low-dose NSAID exposure evaluated for recurrence prevention/reduced metastasis risk	Observational meta-analyses show reduced metastasis/recurrence signals; randomized trials for TNBC prevention/adjunct therapy are limited or ongoing.

## Data Availability

No new data were created or analyzed in this study. Data sharing is not applicable to this article.

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
