# Peer review of "Old Drugs, New Battles: Unleashing Repurposed Drug Classes in Triple-Negative Breast Cancer Treatment"

_ijms, 2025, doi:10.3390/ijms262211196_

Round 1
Reviewer 1 Report
Comments and Suggestions for Authors
The manuscript by Tshimweneka and Mhlanga is a review of the recent examples describing repurposing drugs approved for other conditions to the treatment of TNBC. The review highlights several drug classes and outlines the mechanisms relevant for TNBC progression, which are targeted by these drugs. The information is laid out clearly and concisely. Illustrations are very helpful in supplementing the text.
A few typos, incorrectly phrased sentences, and citation inserts need to be corrected. Some instances may be found in lines 60, 73, 142, 408, 462.
Reviewer 2 Report
Comments and Suggestions for Authors
In the present manuscript, the authors synthesized information from existing review articles and emphasized the benefits and challenges of using FDA-approved repurposed drugs for the treatment of triple-negative breast cancers. The review is well written, and the Figures are helpful but incomplete.
Comments:
Some of the figures take lot of space but convey limited information. It may be beneficial to include a Table that lists repurposed drugs that are used for the treatment of TNBC, drugs used in combination therapy, mechanisms by which the repurposed drugs exert anticancer effects and clinical success.
Figure 4: Although phosphorylation of AMPK activates it, AMPK is activated by low ATP and high AMP. Thus, the arrow pointing ATP towards p-AMPK is somewhat misleading. In addition, the Figures should include additional mechanisms by which metformin exerts anticancer effects in TNBC.
Line 232: Figure 4 should be Figure 5.
Line 233 and Figure 5: While it was mentioned that itraconazole is a potent inhibitor of Hedgehog signaling, the Figure shows it inhibits PI3K downstream of integrin signaling. Figure 5 should also include other potential targets of ketoconazole and itraconazole.
Figure 6: Figure 6 should also include the effects of mebendazole on Wnt/beta-catenin signaling crucial for TNBC stemness and metastasis.
Figure 7: Various mechanisms by which beta-blockers, ACE inhibitors and angiotensin II receptor blockers function could be combined in the same figure.
References listed have 2 numbers and they don’t match after Ref. 22.
Line 540 & 541: The references should be corrected.
The following references should be formatted properly.
Line 16: Ref.14 & 10
Line 142: Ref. 6
Line 408: Ref. 64
Line 462: Ref 65
Line 68: Delete (figure 2). It is already mentioned and more appropriate in line 70.
Line 72 and 73: PARP inhibitors and genotoxic agents.
Reviewer 3 Report
Comments and Suggestions for Authors
The manuscript presents a detailed description of drugs that could be repurposed for treating TNBC. The drugs are presented according to their original use (antidiabetic, antifungal, antiparasitic, antidepressant, cardiovascular, NSAIDs). The manuscript is well written and offers interesting information with useful illustrations.
The following issues could be addressed to improve the quality of the manuscript:
1) The addition of a section describing bioinformatics methods used for identifying which drugs have a potential to be repurposed for treating TNBC. The following reviews offer an overview of this important topic which is not presented in the current version of the manuscript.
https://pmc.ncbi.nlm.nih.gov/articles/PMC10315146/
https://pmc.ncbi.nlm.nih.gov/articles/PMC12471285/
https://www.sciencedirect.com/science/article/pii/S1532046423000941
2) A table presenting completed and ongoing clinical trials with drugs mentioned in the text for treating TNBC could highlight the most promising therapeutic advances.
3) In figure 4 the reduction of ATP levels should be illustrated with an arrow pointing down next to ATP. The addition of a ratio AMP/ATP with an arrow pointing up will illustrate its increase. Then an induction arrow connecting AMP/ATP with p-AMPK will clearly illustrate its activation. The current figure is misleading because it shows that ATP activates p-AMPK and that is not true.
4) In figure 5 the pro-MMP 9 and MMP 9 should be shown on the extracellular space since they are secreted molecules
Reviewer 4 Report
Comments and Suggestions for Authors
Please find the attachment

Round 2
Reviewer 2 Report
Comments and Suggestions for Authors
The authors have satisfactorily responded to prior criticisms. The manuscript is much more improved.
Author Response
Comment 1: The authors have satisfactorily responded to prior criticisms. The manuscript is much more improved.' response to this comment.
Response 1: We thank the reviewer for the positive feedback and are glad the revisions have improved the manuscript.
Reviewer 3 Report
Comments and Suggestions for Authors
The revised manuscript has adressed my main suggestions for improvement.
I would suggest adding a few sentences and 3 references in the introduction about the computational methods used for identifying drugs appropriate for repurposing. This is not an absolute requirement and I do not need to review a revised version of the manuscript with por without this addition.
Author Response
Comment 1: I would suggest adding a few sentences and 3 references in the introduction about the
computational methods used for identifying drugs appropriate for repurposing. This is not an absolute requirement and I do not need to review a revised version of the manuscript with or without this addition.
Response 1: As suggested by the reviewer, information about computational approaches used for identifying drugs appropriate for repurposing has been incorporated, This addition outlines structure-based, signature-based, network-based, and machine learning–driven methods, supported by four references, to highlight the growing significance of computational tools in enabling efficient and precise drug repurposing for cancer therapy. (See page 3 – 4, line 108 – 123 of the revised review article). The added references are on page 18 line 546 – 551.
